# Anillin propels myosin-independent constriction of actin rings

Ondřej Kučera [1,5], Valerie Siahaan[1], Daniel Janda[1], Sietske H. Dijkstra[1], Eliška Pilátová[1], Eva Zatecka[1], Stefan Diez [2,3,4], Marcus Braun [1✉] & Zdenek Lansky [1✉]

Constriction of the cytokinetic ring, a circular structure of actin filaments, is an essential step during cell division. Mechanical forces driving the constriction are attributed to myosin motor proteins, which slide actin filaments along each other. However, in multiple organisms, ring constriction has been reported to be myosin independent. How actin rings constrict in the absence of motor activity remains unclear. Here, we demonstrate that anillin, a nonmotor actin crosslinker, indispensable during cytokinesis, autonomously propels the contractility of actin bundles. Anillin generates contractile forces of tens of pico-Newtons to maximise the lengths of overlaps between bundled actin filaments. The contractility is enhanced by actin disassembly. When multiple actin filaments are arranged into a ring, this contractility leads to ring constriction. Our results indicate that passive actin crosslinkers can substitute for the activity of molecular motors to generate contractile forces in a variety of actin networks, including the cytokinetic ring.

[1] Institute of Biotechnology, Czech Academy of Sciences, BIOCEV, Vestec, Prague West, Czechia. [2] B CUBE – Center for Molecular Bioengineering, TU Dresden, Dresden, Germany. [3] Max Planck Institute of Molecular Cell Biology and Genetics, Dresden, Germany. [4] Cluster of Excellence Physics of Life, Technische Universität Dresden, Dresden, Germany. [5] Present address: CytoMorpho Lab, Laboratoire Physiologie Cellulaire & Végétale, Institut de recherche interdisciplinaire de Grenoble, Commissariat à l'énergie atomique et aux énergies alternatives (CEA), Grenoble, France. ✉email: marcus.braun@ibt.cas.cz; zdenek.lansky@ibt.cas.cz

Constriction of the cytokinetic actin contractile ring drives the division of most eukaryotic cells at the end of mitosis and meiosis[1,2]. The contractile ring is composed of bundles of actin filaments overlapping in mixed orientations, non-muscle myosin-II motors, crosslinking proteins and scaffolding proteins[3–5]. For a long time, it has been suggested that ring constriction is driven by myosin-propelled relative sliding of actin filaments, analogous to muscle sarcomere contraction[6,7]. Unlike muscle sarcomeres, however, the orientation of actin filaments forming the ring is disordered[8]. The sliding activity of myosin alone is thus equally likely to locally lead to contraction or extension, rendering this mechanisms insufficient to generate any net constriction of the ring[9]. Additional factors are therefore required to locally break the symmetry of the system in order to favour contractile forces[3,9–11]. Interestingly, phases of the cytokinetic ring constriction have been shown to be myosin-II-independent. For example in *C. elegans* embryos, constriction continues after conditional inactivation of myosin-II[12] or in *Drosophila* embryos a myosin-II-independent phase of the ring closure was identified during cleavage[13]. Moreover, multiple organisms lack myosin-II completely[14]. Theoretical and experimental works suggest that possible sources of the driving force underlying myosin-independent constriction mechanisms could depend on actin-crosslinking proteins and actin filament disassembly[15–21]. Direct experimental evidence, however, is scarce, leaving the myosin-independent mechanism of actin ring constriction unclear.

Here we show in a minimal reconstituted system that constriction of actin rings can be solely propelled by anillin, a non-motor actin crosslinking and scaffolding protein highly enriched in the contractile ring during cytokinesis[22–24]. Anillin, demonstrated to be required for the completion of cytokinesis in diverse organisms[25–29], is implicated in tumour growth and metastasis[30]. We found that anillin autonomously drives relative sliding of actin filaments and couples with actin filament disassembly to generate contractility. We thus demonstrate that diffusible filament crosslinkers, such as anillin, can generate contractile forces in actin networks, substituting for molecular motor activity.

## Results

### Anillin slides actin filaments to maximise their overlap.

To study the interactions between anillin and actin filaments in vitro, we specifically immobilised sparsely rhodamine-labelled, phalloidin-stabilised actin filaments to the coverslip surface (Methods, Fig. 1a). After the addition of GFP-labelled anillin (Fig. 1b) to the experimental chamber, using TIRF microscopy, we observed anillin-GFP binding to actin filaments (Fig. 1c–e). At an anillin-GFP concentration of 0.12 nM, actin filaments were decorated by anillin-GFP molecules (Fig. 1a, f), which in accordance with previously published data[23,31] we identified as monomers (Fig. S1a). These single anillin-GFP molecules diffused along the actin filaments with a diffusion constant of $0.0088 \pm 0.0006 \ \mu m^2 s^{-1}$ (linear fit coefficient ±95% confidence bounds, 268 molecules in two experiments) (Fig. S1b). When we increased the anillin-GFP concentration in solution to 12 nM and simultaneously added brightly rhodamine-labelled, phalloidin-stabilised actin filaments (mobile filaments), we observed mobile filaments landing from solution and length-wise crosslinking with the immobilised filaments forming filament bundles. In these bundles, we found mobile filaments moving diffusively along the immobilised filaments (Fig. 1g, h, Movie 1), showing that anillin-GFP generates a diffusible link between actin filaments. Strikingly, when a mobile filament landed such that it overlapped partially with the immobilised filament, we observed these mobile filaments moving unidirectionally along the immobilised filaments.

Importantly, this movement was always in the direction increasing the length of the overlap between the two filaments and thus effectively contracting the elementary two-filament actin bundle (Fig. 1i, j, Movie 2). We performed this experiment also in an alternative way, not immobilising any of the filaments, but imaging the bundle formation in solution in the presence of 0.1% (w/v) methylcellulose to facilitate the imaging at a plane near the coverslip surface (Methods). Also, in this set of experiments, performed at 12 or 120 nM anillin concentrations, we observed, after initial formation of a partially overlapping two-filament bundle, relative directional sliding of the two crosslinked filaments increasing their overlap length (Fig. 1k, l, Fig. S1d, and Movie 3). The presence of anillin was required for the formation of the actin bundles and their directional sliding (Fig. S1e and Movie 4). In all experiments, we neither observed directional sliding of filaments that overlapped fully nor directional sliding decreasing the overlap length suggesting that the observed sliding is not due to residual traces of myosin motor protein in the anillin preparation (Supplementary Table 2, Methods). Blocking the motor ATPase activity by replacing ATP with the non-hydrolysable ATP analogue AMP-PNP (β,γ-Imidoadenosine 5′-triphosphate lithium salt hydrate) in presence of hexokinase did not affect the observed sliding of partially overlapping filaments in the presence of anillin-GFP (Fig. S1f, i, Methods), ruling out motor proteins as a possible source of the driving force. Furthermore, directional sliding (increasing the overlap length) did not depend on the relative orientation (structural polarity) of the bundled filaments (Fig. S1g, h), further excluding molecular motors as a source of the movements observed. We furthermore note that the presence of anillin-GFP in our experiments did not alter the length of the actin filaments (Fig. 1k, l and Fig. S1c), excluding filament dynamics as source of the movements observed. Analysing the dynamics of the sliding, we found that the overlap length started increasing immediately after the formation of the bundle. Concomitantly with the increase of the overlap length, the amount of the anillin-GFP in the overlap increased (Fig. S1j), indicating anillin-GFP binding into the newly forming overlap region. Interestingly, sliding slowed down as the overlap length and the amount of anillin in the overlap increased, leading to the overlap length reaching an equilibrium value (Fig. 1m, n and Fig. S1k, l). Combined, these data demonstrate that anillin drives directional sliding of actin filaments relative to each other, increasing the length of the overlap between the filaments and, thus, contracting the filament bundle.

### Anillin generates tens of pico-Newton forces to slide actin filaments.

To directly measure the forces generated by anillin in the actin bundles, we specifically attached actin filaments to two microspheres (Methods). Holding the microspheres by two traps in an optical tweezers setup, we formed a crosslinked actin bundle consisting on average of about five filaments (4.7 ± 1.3, mean ± s.d., n = 35 bundles, Methods) in the presence of 12 nM anillin-GFP in solution, (Fig. 2a, b, Methods, and Fig. S2a). After ~5 min, when the system reached equilibrium, we started moving the microspheres apart from each other in 100 nm steps, stretching the bundle and thus sliding the filaments within the bundle in the direction of shortening their overlaps (Fig. 2b). Simultaneously with each step-movement of the trap (each step taking ~33 μs), we observed a rapid increase in force, which then decayed within several seconds to a plateau reflecting the equilibrium between the anillin-generated force, exerted in the direction of increasing the lengths of the overlaps between the filaments in the bundle, and the external load exerted by the optical trap in the opposite direction (Fig. 2c, d, see Fig. S2b for control experiment). Since the median time constant of the force decay (1.06 s, quartiles

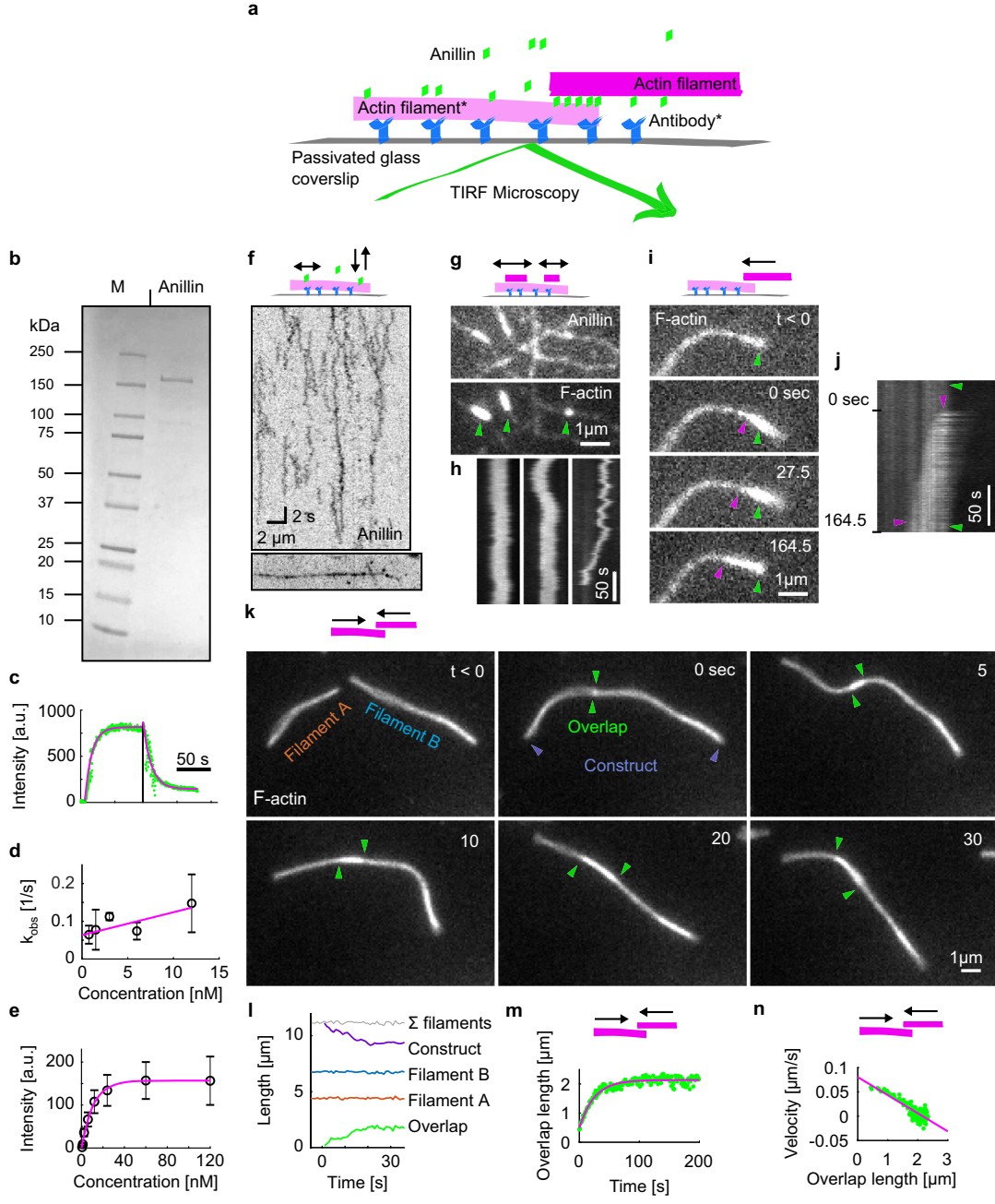

0.47 s and 2.00 s, $n = 326$ steps in 43 experiments) together with the trap movement was an order of magnitude faster than the rate of anillin unbinding from actin (Fig. 1d, Methods), we interpret this dynamic response being due to the rearrangement of anillin molecules in the overlap. The equilibrium force values increased with increasing distance between the two microspheres and thus with decreasing the overlap lengths between the filaments in the bundle ($n = 326$ steps in 43 experiments, Fig. 2c, e). On the timescale of the experiment, the decrease in the lengths of the overlaps resulted in an increase in the density of anillin-GFP in the overlaps (Fig. 2e, inset). The equilibrium force values reached up to tens of pico-Newtons before dropping suddenly, presumably when an actin filament slid apart completely from the rest of the bundle (Fig. S2c, d). When we (before the filament slid apart completely, in a situation when the bundle was pre-stretched) decreased the external load on the bundle by moving the

microspheres closer together in steps of 100 nm, we immediately observed a decrease of the force, highlighting the reversibility of the process (Fig. 2d). We thus conclude that at constant anillin concentration, anillin molecules crosslinking actin filaments can autonomously generate substantial forces, which increase with decreasing lengths of the overlaps between the filaments and thus with increasing density of anillin crosslinkers in the overlap.

To test how the generated force depends on changes in the anillin concentration, we used the setup described above with an actin-anillin bundle suspended between two microspheres. We first pre-stretched the actin-anillin bundle by moving the microspheres apart such that the force readout was nonzero, and then we decreased the concentration of anillin-GFP in solution from 12 to 1 nM while keeping the distance between the microspheres constant (Methods). We then observed the force generated by anillin decreasing over time, presumably as

**Fig. 1 Anillin slides actin filaments to maximise their overlap. a** Schematic representation of the experimental setup. The asterisk denotes components used only in experiments with immobilised filaments. **b** SDS gel of anillin-GFP used in this study. This experiment was repeated three times with similar results. **c**, Kinetics of the anillin-GFP binding/unbinding to/from actin filaments. A temporal profile of the fluorescence signal of the anillin-GFP along an actin filament shows a typical first-order growth upon addition of 6 nM anillin-GFP and first-order decay after the washout. The loading experiment was repeated 15 times (three experiments) with similar result. **d** Concentration profile of the observed kinetics of anillin-GFP binding. The linear fit to the mean values ($R^2 = 0.64$) was used to estimate the unbinding rate of anillin from actin filaments, $k_{off} = 0.06\,s^{-1}$, 95% confidence interval 0.03–0.10 $s^{-1}$ ($n = 15$ measurements in three experiments). Error bars represent mean ± s.d. **e** Concentration profile of the steady-state density of anillin-GFP on actin filaments (right). Michaelis–Menten fit to the mean values ($R^2 = 0.98$) was used to estimate the dissociation constant, $K_d = 7.82\,nM$, 95% confidence interval 5.53–10.11 ($n = 12$–36 measurements per concentration in three experiments). Error bars represent mean ± s.d. **f** Intensity-inverted kymograph showing single anillin-GFP molecules diffusing along an actin filament (filament shown below the kymograph). **g** Fluorescence micrographs showing three short mobile actin filaments (indicated by green arrowheads) crosslinked by anillin-GFP to long, sparsely fluorescently labelled, and immobilised actin filaments (see Movie 1). **h** Kymographs show the diffusion of the mobile actin filaments from the micrograph in **g** along the immobilised filaments (see Movie 1). **i** Time-lapse micrographs and a kymograph, **j**, showing anillin-driven sliding of a mobile actin filament (bright) along an immobilised filament (dim), increasing the overlap between the two actin filaments (see Movie 2). Green and magenta arrowheads indicate the ends of the immobilised and mobile filaments, respectively. This experiment was repeated five times (eight events observed) with similar results. **k** Time-lapse micrographs showing anillin-driven sliding of two mobile actin filaments along each other, increasing the overlap between the two actin filaments (see Movie 3). Arrowheads indicate the ends of the filaments. This experiment was repeated 15 times (24 events observed) with similar results. **l** Time-traces of the lengths of the two filaments, the length of the overlap and the length of the bundle construct shown in the micrographs in **k**. The bundle contracts as the overlap length increases, while the lengths of the two filaments are constant. **m** A typical time trace of the overlap expansion (green dots) reaching an equilibrium value (magenta line represents an exponential fit (Methods) to the data). **n** Velocity of the overlap expansion decreases with increasing overlap length. Green points represent an exemplary event, magenta line is a linear fit to the data.

anillin-GFP was unbinding from the overlap (Fig. 2f and see Fig. S2f for control experiment). Conversely, increasing the anillin-GFP concentration in the measurement chamber from 1 to 12 nM while keeping the distance between the microspheres constant resulted in a gradual increase in the anillin-generated force, presumably due to anillin-GFP binding into the overlap (Fig. 2g and Fig. S2g).

Combined these experiments show that at constant anillin concentration, anillin molecules in the overlap generate force, which is inversely proportional to the length of the overlap between the filaments and thus proportional to the density of anillin crosslinkers in the overlap. Additional increase or decrease in the concentration of anillin results in an additional increase or decrease of the generated force, respectively. Importantly, these forces are always directed such that they increase the overlap lengths between actin filaments in the bundle and, consequently, result in the contraction of the filament bundle.

**Anillin couples with actin disassembly to generate directed filament sliding.** As actin depolymerisation is an important factor for the constriction of the cytokinetic ring, we wondered how depolymerisation of actin filaments will affect the observed anillin-driven filament sliding. We employed the experimental setup described in Fig. 1, using a mixture of Atto647-labelled, phalloidin-stabilised actin filaments and rhodamine-labelled non-stabilised actin filaments in the presence of 12 nM anillin-GFP and not immobilising any of the filaments. To enhance the disassembly of the non-stabilised filaments, we used 80–200 nM latrunculin A[32] leading to the non-stabilised actin filaments disassembling at a rate of $1.2 \pm 1.1$ monomer/s, (mean ± s.d., $n = 13$). Similar to the experiment in Fig. 1, we observed anillin-dependent crosslinking of the filaments leading to the formation of actin bundles, which contracted over time. Strikingly, however, in this experiment, we did not observe only sliding of partially overlapping filaments as described above, but we also observed sliding of filaments that overlapped fully. This sliding was observed when the retreating tip of a disassembling non-stabilised actin filament reached the end of a stabilised filament crosslinked to the disassembling filament by anillin (Fig. S3 and Movie 5). The stabilised filament then followed the retreating tip of the non-stabilised actin filament (Fig. 3a, b and Movies 6, 7) such that the velocity of its sliding matched the rate of filament disassembly

(correlation coefficient $0.90 \pm 0.06$, mean ± s.d., $n = 13$ events in nine experiments) (Fig. 3c–e). Consequently, full overlaps were maintained between the stabilised filaments and the disassembling filaments to which they were crosslinked (Fig. 3a, b and Fig. S3). We hypothesise that as the disassembly of one of the filaments transiently shortens the overlap length, the concurrent anillin-driven sliding described in the experiments presented in Fig. 1 compensates this shortening, maintaining the overlap length. This process, leading to the sliding of actin filaments along with the tip of a depolymerising filament, is thus an efficient mechanism that converts actin filament depolymerisation into sliding-driven contraction of actin bundles.

**Anillin promotes the formation of actin rings and drives their constriction.** To study the anillin-driven sliding of actin filaments in networks of higher complexity, we increased the actin filament density in solution, while not immobilising any of the filaments. Initially, we used non-stabilised actin filaments. We found that anillin, similar as septins[33], promotes the formation of actin-filament rings with several micrometres in diameter (Fig. 4a, b and Fig. S4a, b, Methods). We note that further increasing the density of actin filaments in solution resulted in a formation of dense actin networks (Fig. S5). The rings, which we observed throughout the volume of the microscopy chamber, typically consisted of three to eight actin filaments colocalising with anillin-GFP (Methods). Importantly, time-lapse imaging of these rings revealed that they constrict over time (Fig. 4b–d and Movie 8). The circumference of the rings decreased asymptotically (Fig. 4c–e), enabling us to quantify the circumference of the maximally constricted ring. The constriction varied substantially, with constriction after 300 s reaching up to 20% of the initial circumference, maximum asymptotical constriction reaching 100 % and the maximum constriction rate reaching 0.034 $\mu m\,s^{-1}$ (Fig. 4f, h and Fig. S4c). The spread in the data likely reflects the spread in the initial lengths of the overlaps between the filaments forming the rings, as well as the spread in the number of these filaments. While most of the rings ($n = 18$, out of total 26 rings in 16 experiments) constricted during the time of the experiment, in some cases ($n = 8$), we did not observe any significant constriction. We interpret these as rings that already reached their maximum constriction before the beginning of our observation. To test the contribution of actin disassembly in the constriction process, we additionally studied the constriction

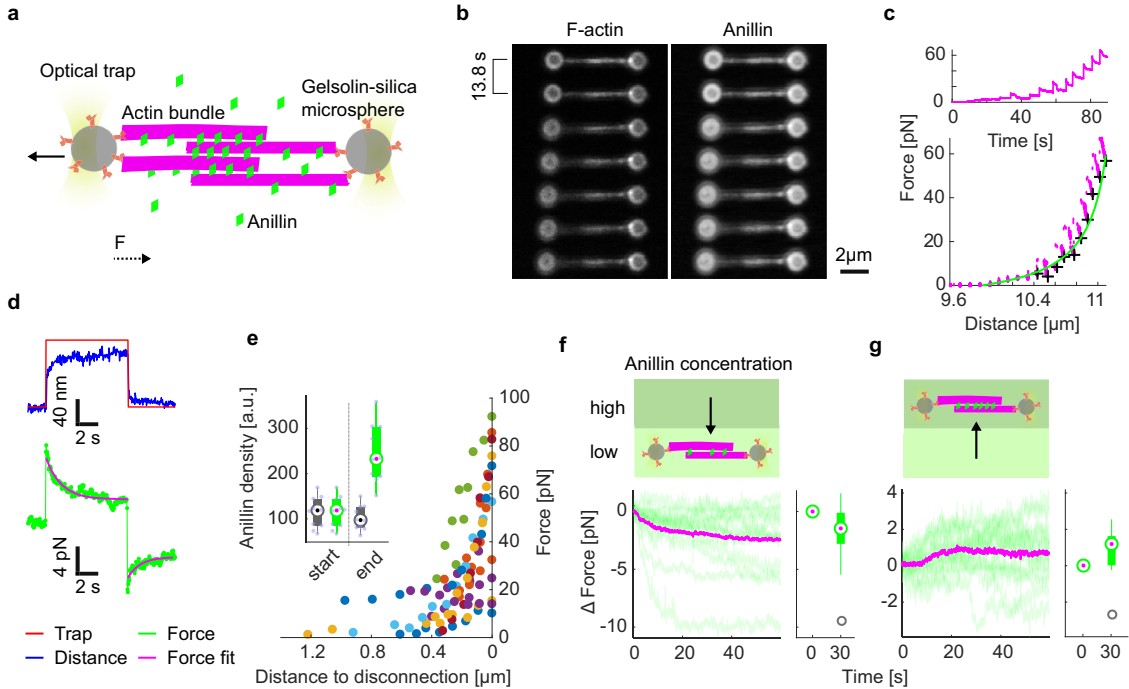

**Fig. 2 Anillin generates tens of pico-Newton forces to slide actin filaments. a** Schematic representation of the experimental setup. **b** Time-lapse fluorescence micrographs showing an actin bundle attached between two silica microspheres. The bundle is being stretched as the left microsphere is pulled leftwards by an optical trap. **c** Typical force time-trace (top) and the force-distance curve (bottom) corresponding to stretching of an anillin-actin filament bundle (experimental data points—magenta). Asymptotic forces of individual stretching steps, calculated by fitting an exponential to the force decays (as shown in **d**), are indicated by black crosses. These increase hyperbolically with increasing distance between the microspheres, and thus with decreasing overlap length $L$. The green line represents ~$1/L$ fit to the data. **d** Temporal response of the construct to stretching and relaxation; the left optical trap is moved 100 nm away from the right trap and then, after ~7 s, moved back to the original position. The temporal profile of the longitudinal position of the left optical trap is shown together with the detected distance between the microspheres (top) and the detected force (bottom). **e** The detected force increased with decreasing overlap length before the filaments slid apart completely (distance to disconnection = 0). All events longer than six steps are plotted ($n = 11$ experiments indicated by different colours). Inset, anillin density in the overlap (fluorescence intensity of anillin per unit length of the overlap) at the start and the end of the bundle stretching. Grey boxplots represent raw data, green boxplots represent data after photobleaching correction (see Fig. S2e for the photobleaching estimation) ($n = 10$ experiments). Corresponding data points overlay the boxplots. **f**, **g** Force response of a pre-stretched actin-anillin bundle to a decrease (**f**) or increase (**g**) of anillin-GFP concentration. Schematic representation of the experiment (top) and temporal experimental data (bottom). Decrease of the concentration ($n = 15$ events in 14 experiments), increase of the concentration ($n = 15$ events in 14 experiments). Green curves are the experimental data, mean temporal profile is shown in magenta. Box and whisker plots show a significant decrease or increase in force between time points 0 and 30 s after a decrease (one-sided Wilcoxon test, $p = 0.02$) or increase (one-sided Wilcoxon test, $p = 0.03$) of anillin-GFP concentration. In **e**–**g** data were represented as boxplots. Central marks represent median, top and bottom edges of the box indicate the 75th and 25th percentiles, respectively. Whiskers extend the most extreme points that are not considered outliers. Outliers are marked as grey circles.

dynamics of anillin-actin rings consisting of filaments stabilised by rhodamine-phalloidin. We found that the constriction was significantly slower ($P = 0.0044$), with constriction after 300 s of up to about 5% of the initial circumference and maximum constriction rate of $0.006 \, \mu m \, s^{-1}$ (Fig. 4f–h, $n = 12$ rings in ten experiments). However, the maximum asymptotical constriction reached 100%, similarly to the rings formed by disassembling actin filaments. Consistently, the addition of latrunculin to rings of non-stabilised actin filaments led to higher asymptotic constriction (Fig. 4g, h). These results are in agreement with our data on linear bundles, where we observed that depolymerisation of actin filaments promotes the contraction of the bundle. Combined these experiments show that rings of actin filaments can constrict without the action of nucleotide hydrolysing molecular motors, solely driven by forces generated by passive actin crosslinkers. This constriction is enhanced by actin-filament disassembly.

## Discussion

We here show that the passive, non-motor, actin crosslinker anillin can drive the contraction of actin bundles through directed filament sliding. Depletion interactions[34–37] did not contribute to

this phenomenon, as filaments did neither bundle nor slide in the absence of anillin in our control experiments (Fig. S1e). Actin filaments were sliding also in the absence of ATP, ruling out the remnants of myosin in our anillin-GFP preparation as the possible sources of sliding. Importantly, we observed only sliding of partially overlapping filaments and the sliding was always directed such that the length of the overlap between the filaments increased; both findings ruling out molecular motors as possible source of the movement. Anillin diffuses along actin filaments (Fig. 1f), creating a diffusible link between actin filaments (Fig. 1g, h), and is retained in overlaps that transiently shorten (Fig. 2e, inset), suggesting that anillin molecules in the overlaps can be described as particles of an ideal gas confined in the overlap, analogous to diffusible crosslinkers in microtubule systems[38,39]. We observed that the anillin-generated force is inversely proportional to the overlap length (Fig. 2c, e) and, considering that anillin unbinding on the experimental timescale was negligible, thus proportional to the density of anillin in the overlap, as expected for an ideal gas[38]. Increasing or decreasing the concentration of anillin in our trapping experiment while keeping the distance between the traps constant, resulted in an increase or

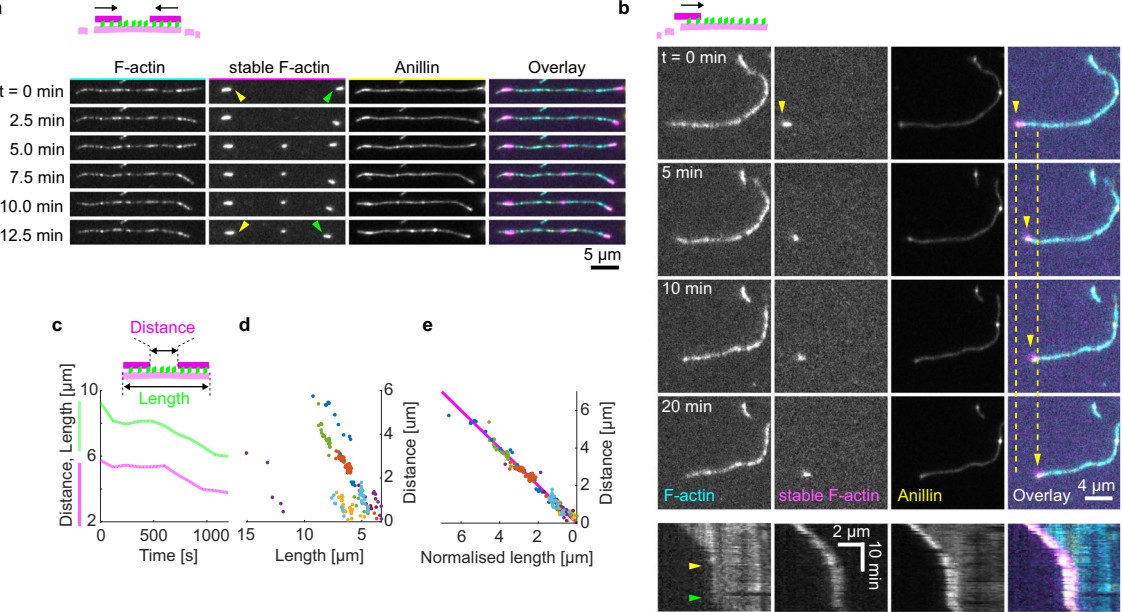

**Fig. 3 Anillin couples with actin disassembly to generate directed filament sliding. a** Fluorescence time-lapse micrographs showing two short phalloidin-stabilised actin filaments sliding along with the retreating ends of a non-stabilised disassembling actin filament (see Movie 6). Arrowheads indicate the inner ends of the two stabilised actin filaments coming closer to each other. **b** Fluorescence time-lapse micrographs showing one short phalloidin-stabilised actin filament sliding along with the retreating end of a non-stabilised disassembling actin filament. Below, multicolour fluorescence kymograph of the event, showing that transient pauses in the disassembly (beginning and end of the pause indicated by yellow and green arrowheads) result in transient pauses in the filament sliding (see Movie 7). **c** Typical time-traces of the length of the non-stabilised actin filament (green) and the distance between the two short stabilised filaments (magenta) moving with the ends of the non-stabilised actin filament. **d,e** The distance between the stabilised actin filaments decreases linearly with the length of the disassembling actin filament ($n = 12$ events in nine experiments, colours denote individual events).

decrease of the anillin-generated force, respectively. Using the analogy of the ideal gas, we interpret this observation as an increase or decrease in the anillin 'pressure' in the overlap after increasing or decreasing the number of particles in a constant volume. Combined, our results thus suggest that the anillin-generated force is of an entropic origin[40] (Fig. 5a). The ability of monomeric anillin (Fig. S1a) to bundle actin filaments (Fig. 1), and to generate entropic force in partially overlapping actin bundles (Fig. 2), apparently originates from its possession of multiple actin-binding sites[41], highlighting the importance of multivalency for the process[42]. An additional force component is likely associated with energetically favourable binding of the anillin crosslinkers into the overlap. At anillin concentrations of 12 nM used in this study, anillin molecules, collectively, can generate forces in the order of 10 pN in a bundle of about five filaments. These forces are comparable to forces generated by myosins; single Myosin-II stall force is about 2 pN[43,44], suggesting that anillin crosslinkers can generate forces relevant in the context of the cytokinetic ring.

Motion of filament-associated proteins along the surface of cytoskeletal filaments induces frictional forces[45]. During the relative sliding of actin filaments, anillin molecules have to collectively reorganise within the overlap, likely moving along the surfaces of the two filaments, which will thus result in the generation of frictional forces acting against the forces driving the movement. Frictional forces increase with the increasing number of crosslinkers coupling the two filaments, and with the crosslinker density[38,39,46–48]. The generation of friction can thus explain the observed slowdown in filament sliding concomitant with the increase of anillin molecules in the overlap when the overlap length increases (Fig. 1m, n). Similarly, friction will likely play a role in the observed slowdown of the ring constriction (Fig. 3c–f). Circular constraints within the ring may also contribute to the deceleration of the constriction as the decrease of

the diameter of the ring requires stronger bending of its constituting filaments[33,37,49]. Dependent on the crosslinker stiffness and their interaction potential, the crosslinker induced friction generated between sliding filaments can exhibit highly non-linear scaling with the crosslinker number and density[46]. As the entropic force driving the contraction scales linearly with the crosslinker density, friction might become prohibitive at high anillin concentrations. Although we could not test the mechanism at elevated anillin concentrations due to technical reasons, these considerations suggest the existence of an anillin concentration optimal for the constriction. While below this concentration the bundling of the filaments would be inefficient, and the contractile forces low, above this concentration the frictional forces might become prohibitive and the filament sliding would stall. This notion is in accordance with in vivo observations showing that intermediate levels of anillin enable maximum speed of contraction of the cytokinetic ring in *C. elegans* zygote[17].

Anillin is required for the ingression of the cytokinetic furrow and the completion of cytokinesis[26,29,50]. Depletion of anillin leads to slowdown of the ingression and cytokinesis failure[27,28]. In this context anillin is thought of mostly as a scaffolding element and a regulator of myosin-driven contractility. However, our experiments show that anillin is an autonomous generator of contractile forces. Furthermore, actin filament depolymerisation in various, albeit not all, organisms[51] plays an important role during the constriction of the cytokinetic ring. Blocking actin depolymerisation, e.g. with drugs or by impairing the actin-depolymerising and severing enzyme cofilin, leads to defective ring constriction[21,52–54]. Moreover, decreased levels of cofilin or anillin resulted in similar effect in *Drosophila* embryos, namely a delayed switch to the myosin-independent phase of the constriction[13], suggesting that actin filament depolymerisation, in concert with anillin might drive myosin-independent constriction of the cytokinetic ring. We here provide direct evidence that anillin crosslinkers efficiently couple

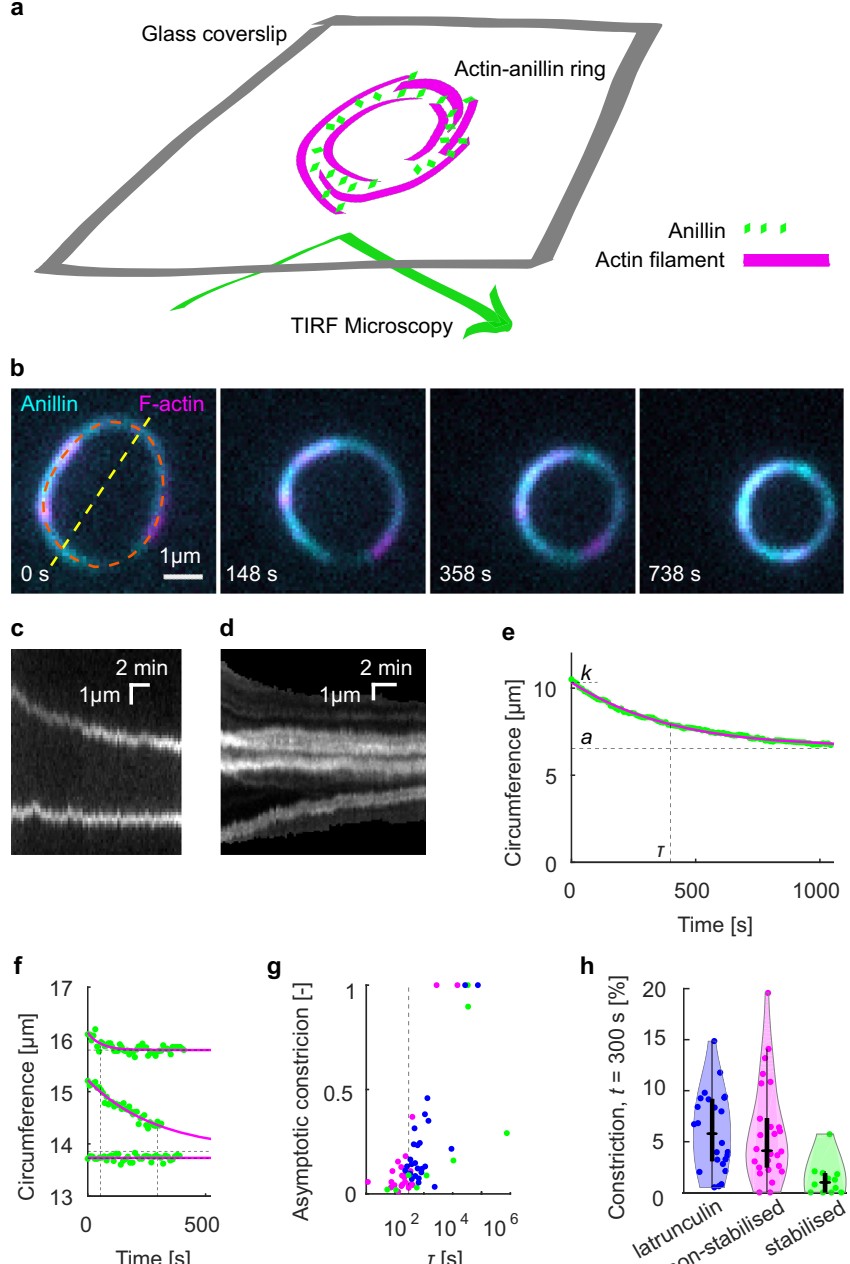

**Fig. 4 Anillin promotes the formation of actin rings and drives their constriction. a** Schematic representation of the assay geometry. **b** Multicolour time-lapse micrographs of the anillin-actin ring, showing the ring constriction over time (see Movie 8). **c** Kymograph along the yellow dashed line (diameter) in **b** of the constricting anillin-actin ring. **d** Kymograph along the orange dashed line (circumference) in **b** of the constricting anillin-actin ring showing the relative sliding of fluorescently sparsely labelled (speckled) filaments, which form the ring. Relative sliding is visualised by the relative motion of fluorescent speckles. **e** Time trace of the circumference of the ring shown in **b**. Green dots represent experimental data and the solid magenta curve an exponential fit. Dashed lines indicate the fit parameters ($\tau$ time constant, $k$ initial circumference, $a$ asymptotical circumference). **f** Exemplary time-traces of fast (top), slow (centre) and not (bottom) constricting rings. Identical graphical representation as in **e**. See Fig. S4c for the time traces of all events. **g** Relation between the time constant and the relative asymptotic constriction ($1 − (a/k)$). Each dot represents one ring formed either by non-stabilised (magenta, $n = 26$ in 16 experiments) or stabilised (green, $n = 12$ in 10 experiments) actin filaments, or non-stabilised actin filaments in the presence of latrunculin (blue, $n = 25$ in 7 experiments). Fits leading to asymptotical constriction higher than 1 were cut off to 1 ($n = 5$). Events with higher time constants than $10^6$ s, all having zero asymptotic constriction, are not shown ($n = 5$). $\tau = 300$ s is highlighted by vertical dashed line. **h** Percentage of the constriction at $t = 300$ s for all observed rings of stabilised and non-stabilised actin filaments, or non-stabilised actin filaments in the presence of latrunculin ($n = 63$). Individual data points are accompanied by violin plots, and black boxplots: Central marks represent median, top and bottom edges of the box indicate the 75th and 25th percentiles, respectively. Whiskers extend the 95% confidence intervals.

with actin filament disassembly to generate contractile forces (Fig. 5b, c). This mechanism is reminiscent of the experimentally observed mechanism driving chromosome movement during the anaphase of cell division by coupling the chromosomes' kinetochores to depolymerising microtubule ends by biased diffusion[55–57] as proposed earlier theoretically[58]. Analogously, the diffusion of anillin is likely biased away from depolymerising actin filament ends, resulting in directional sliding of crosslinked filaments with the depolymerising end. We hypothesise that severing of actin filaments might promote this process, by increasing the

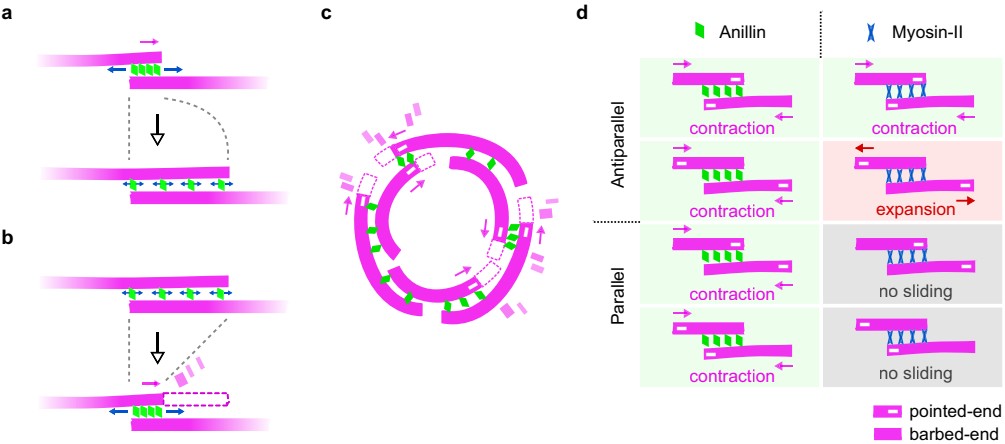

**Fig. 5 Mechanism and properties of anillin-driven actin contractility. a** Entropic expansion of anillin molecules confined between two partially overlapping actin filaments leads to filament sliding which maximises the length of the overlap and contracts the bundle (top). Sliding ceases when the entropic forces come to an equilibrium with forces opposing the sliding, such as frictional forces between the filaments (bottom). **b** Filament disassembly (for clarity, only one filament is shown to disassemble) transiently shortens the overlap, resulting in further sliding through entropic anillin expansion, again maximising the overlap length. **c** When actin filaments are organised in a ring geometry with mixed actin-filament polarities, anillin-propelled contraction of actin bundles will lead to ring constriction. **d** Comparison of actin contractility driven by diffusible crosslinkers such as anillin (left column) and molecular motors such as myosin (right column). While the entropic expansion of the crosslinkers in the overlap always leads to contraction, irrespective of the relative orientation of the actin filaments (filament polarities indicated by —), the action of motors is governed by the polarity of the filaments[11] and may result in contraction, extension or no sliding.

number of filament ends, which can trigger at these positions additional anillin-mediated filament sliding and accelerate the bundle contraction.

Molecular motor-driven sliding of filaments depends on the relative polarities of the filaments in the bundle[11] (Fig. 5d). In a disordered actin array, such as in the cytokinetic ring, myosin motors alone are thus equally likely to locally generate contractile or extensile forces. Additional mechanisms are needed to break the symmetry of the process, locally favouring the contractile forces to drive the overall net constriction of the ring. In this context, various factors are currently under debate (reviewed extensively in refs. [3,10,11]), such as the difference in behaviour of actin filaments under compressive and extensile forces[59,60], inhomogeneous distribution of myosin motors[19,61–63] or membrane anchoring of actin filaments[44]. Importantly, connectivity between actin filaments mediated by actin crosslinker α-actinin facilitates myosin-driven contraction of actin rings in vitro[9]. We here show that anillin-generated forces always act in the direction increasing the overlap length, contracting the actin bundle. Anillin will thus generate assisting forces in overlaps, which are locally contracted by myosin and resisting forces in overlaps, which are locally extended by myosin. Therefore, anillin-generated forces might constitute an element breaking the symmetry for myosin-dependent force generation in the cytokinetic ring and thus, in concert with myosin, drive its constriction. This notion is consistent with in vivo results showing that, in *Drosophila* embryos, lack of anillin results in a complete stall of myosin-driven contraction in the myosin-dependent phase of the ring constriction[13].

While the previous observations of the entropic cytoskeletal contractility were limited to two cytoskeletal filaments, here we demonstrate that diffusible cytoskeletal crosslinkers can generate forces in networks of higher complexity, such as bundles or rings, as hypothesised previously[64]. We show directly that actin crosslinkers can drive the contraction of actin filament bundles and can couple with actin filament disassembly to enhance this effect. We propose that this mechanism can (i) underpin the in vivo observed myosin-independent constriction of cytokinetic rings and (ii) break the symmetry of the randomly oriented actin-

myosin system in the ring enabling myosin-dependent constriction of the ring. A recent finding that anillin promotes tensile forces in the apical actomyosin networks of *X. laevis* embryonal epithelium[65] suggests that analogous mechanism might also be employed in other actin structures. We propose that crosslinker-dependent contractility of filamentous networks is a fundamental mechanism readily available in actin-based cytoskeletal structures, likely contributing to various cellular movements.

## Methods

**Proteins.** *Homo sapiens* anillin (GeneBank accession number: BC070066) cDNA was purchased (BC070066-seq-TCHS1003-GVO-TRI, BioCat GmbH Heidelberg, Germany), PCR amplified (for primers used see Supplementary Table 1) and ligated into an AscI-NotI-digested pOCC destination vector[66] containing a C-terminal GFP tag followed by a 3 C PreScission Protease cleavage site and a 6xHis-tag. The protein was expressed in SF9 insect cells using the opensource FlexiBAC baculovirus vector system[66]. The insect cells were harvested after 4 days by centrifugation at 300xg for 10 min at 4 °C in an Avanti J-26S ultracentrifuge (JLA-9.1000 rotor, Beckman Coulter). The cell pellet was resuspended in 5 ml ice-cold phosphate buffered saline (PBS) and stored at −80 °C for further use. For cell lysis, the insect cells were homogenised in 30 ml ice-cold His-Trap buffer (50 mM Na–phosphate buffer, pH 7.5, 5% glycerol, 300 mM KCl, 1 mM MgCl₂, 0.1% Tween-20, 5 mM BME and 0.1 mM ATP) supplemented with 30 mM imidazole, Protease Inhibitor Cocktail (cOmplete, EDTA free, Roche) and benzonase to the final concentration of 25 units/ml, and centrifuged at 45,000 × g for 60 min at 4 °C in the Avanti J-26S ultracentrifuge (JA-30.50Ti rotor, Beckman Coulter). The cleared cell lysate was incubated in a lysis buffer-equilibrated Ni-NTA column (HisPur Ni-NTA Superflow Agarose, Pierce, VWR) for 2 h at 4 °C on a rotator. The Ni-NTA column was washed with the wash buffer (His-Trap buffer supplemented with 60 mM imidazole) and the protein was eluted with the elution buffer (His-Trap buffer supplemented with 300 mM Imidazole). The fractions containing anillin-GFP were pooled, diluted 1:10 in the His-Trap buffer and the purification tag was cleaved overnight with 3 C PreScisson protease. The solution was reloaded onto a Ni-NTA column to further separate the cleaved protein from the 6xHis-tag. The protein was concentrated using an Amicon ultracentrifuge filter and flash-frozen in liquid nitrogen. Protein concentration was measured using Bradford assay (23236, Thermo Scientific) and microplate reader spectrometer (CLAR-IOstar, BMG Labtech, Germany).

The constitutively active (*R. norvegicus*, amino acids 1–430) recombinant C-terminal eGFP-hexa-histidine tagged kinesin-1 was expressed in *E. coli* BL21-CodonPlusVR (DE3)-RIPL (Stratagene) induced with 0.5 mM IPTG for 16 h at 18° C. Harvested cells were resuspended in buffer A (50 mM sodium phosphate buffer pH 7.5, 1 mM MgCl₂, 10 mM 2-mercaptoethanol, 300 mM NaCl, 0.1% Tween-20 w/vol, 10% glycerol w/vol, 30 mM imidazole and EDTA-free protease inhibitors (Roche)) and lysed using an EmulsiFlex high pressure homogeniser (Avestin) at 4 °

C. Crude lysate was centrifuged at 20,000x$g$ at 4 °C and the supernatant was loaded on Ni-NTA resin (Qiagen). The resin was washed with buffer A containing 60 mM imidazole. The protein was eluted in buffer A containing 300 mM imidazole, snap frozen and stored in –80°C.

To determine whether there are residual traces of non-muscle myosin-II in the preparation of anillin-GFP expressed in SF9 insect cells (*S. frugiperda*, taxid 7108), we performed proteomic mass spectrometry analysis of the preparation. To the sample, we added 100 mM TEAB (triethylammonium bicarbonate) containing 2% SDC (sodium deoxycholate), 10 mM TCEP and 50 mM chloroacetamide to the final volume of 100 μl. Samples were lysed, reduced and alkylated in one step at 95 °C for 10 min and digested with 1 μg of trypsin at 37 °C overnight. After digestion, samples were acidified with TFA to 1% final concentration. SDC was removed by extraction to ethylacetate[67], and peptides were desalted using in-house made stage tips packed with C18 discs (Empore), according to Rappsilber[68].

Nano reversed-phase column (EASY-Spray column, 50 cm × 75 μm ID, PepMap C18, 2 μm particles, 100 Å pore size) was used for LC/MS analysis. Mobile phase buffer A was composed of water and 0.1% formic acid. Mobile phase B was composed of acetonitrile and 0.1% formic acid. Samples were loaded onto the trap column (Acclaim PepMap300, C18, 5 μm, 300 Å Wide Pore, 300 μm x 5 mm) at a flow rate of 15 μl/min. Loading buffer was composed of water, 2% acetonitrile and 0.1% trifluoroacetic acid. Peptides were eluted with a gradient of B from 4 to 35% over 60 min at 300 nl/min flow rate. Eluting peptide cations were converted to gas-phase ions by electrospray ionisation and analysed on a Thermo Orbitrap Fusion (Q-OT- qIT, Thermo). Survey scans of peptide precursors from 350 to 1400 m/z were performed at 120 K resolution (200 m/z) with a $5 \times 10^5$ ion count target. Tandem MS was performed by isolation at 1,5 Th with the quadrupole, HCD fragmentation with a normalised collision energy of 30, and rapid scan MS analysis in the ion trap. The MS 2 ion count target was set to $10^4$, and the max injection time was 35 ms. Only those precursors with charge state 2–6 were sampled for MS 2. The dynamic exclusion duration was set to 45 s with a 10 ppm tolerance around the selected precursor and its isotopes. Monoisotopic precursor selection was turned on. The instrument was run in top speed mode with 2 s cycles[69].

All data were analysed and quantified with the MaxQuant software (version 1.6.10.43)[70]. The false discovery rate (FDR) was set to 1% for both proteins and peptides, and we specified a minimum length of seven amino acids. The Andromeda search engine was used for the MS/MS spectra search against the Anillin sequence and database of expression system—*Spodoptera frugiperda* database (downloaded from Uniprot on July 2020, containing 26 645 entries). Precursor mass tolerance was set to 4.5 ppm, and MSMS match tolerance to 0.5 Da. Enzyme specificity was set as C-terminal to Arg and Lys, also allowing cleavage at proline bonds and a maximum of two missed cleavages. Carbamidomethylation of cysteine was selected as fixed modification and N-terminal protein acetylation and methionine oxidation as variable modifications. The 'match between runs' feature of MaxQuant was used to transfer identifications to other LC-MS/MS runs based on their masses and retention time (maximum deviation 0.7 min), and this was also used in quantification experiments. Data analysis was performed using Perseus 1.6.10.43 software[71]. To identify myosin-II among the proteins detected in the preparation, we performed Blast (blastp: protein vs. protein) of the amino acid sequence of *D. melanogaster* non-muscle myosin-II (uniprot Q99323 and flybase FBgn0265434) against the *S. frugiperda* genome and found GSSPFG00012506001-PA to be the protein with the highest identity (80% identities, 89% positives). Blasting the GSSPFG00012506001-PA sequence in UNIPROT against the Arthropoda database gave us the UniProt protein ID number A0A2H1WPK0, which was searched in the mass spectrometry results (please refer to Supplementary Table 2 for the relevant excerpt; the full table is included in the supplementary dataset).

Unlabelled rabbit muscle actin (AKL99, Cytoskeleton, Inc., USA) and rhodamine-labelled rabbit muscle actin (AR05, Cytoskeleton, Inc., USA) was resuspended to final concentration of 10 mg/ml in general actin buffer (GAB; 5 mM TRIS-HCl pH 8, 0.2 mM CaCl₂) supplemented with 0.2 mM ATP, 5% (w/v) sucrose and 1% (w/v) dextran. For experiments with fluorescently labelled non-stabilised filaments, rhodamine-labelled actin was diluted with the unlabelled actin above in a 1:10 ratio. The actin was then aliquoted, flash-frozen and stored in −80 °C. Actin filaments were polymerised by mixing the aliquoted actin (0.32 mg/ml final concentration) with polymerisation buffer (5 mM Tris-HCl pH 8.0, 0.2 mM CaCl₂, 50 mM KCl, 2 mM MgCl₂, 1 mM ATP) and, optionally, phalloidin (15 μM final concentration) for filament stabilisation and additional labelling. Biotinylated phalloidin (biotin-XX phalloidin, B7474, Thermo Fisher Scientific), fluorescence-labelled phalloidin (rhodamine-phalloidin, R415, Thermo Fisher Scientific, or Atto647 phalloidin, 65906, Sigma) or a 3:4 mixture of both were used in the experiments. Phalloidin-stabilised filaments were polymerised overnight at 4 °C and remained stable for at least 2 months. Non-stabilised filaments were grown in the polymerisation solution in the absence of phalloidin for ~30 min. They were employed from 5 min to 5 h after the polymerisation. For use in the assays, actin filaments were diluted in a ratio between 1:10 and 1:300, depending on the experiment.

Fluorescently-labelled microtubules were polymerised from 4 mg/ml porcine tubulin (80% unlabelled and 20% Alexa Fluor 647 NHS ester-labelled; Thermo Fisher Scientific) for 2 h at 37 °C in BRB80 (80 mM PIPES, 1 mM EGTA, 1 mM MgCl₂, pH 6.9) supplemented with 1 mM MgCl₂ and 1 mM GMPCPP

(Jena Bioscience, Jena, Germany). The polymerised microtubules were centrifuged for 30 min at 18,000 x $g$ in a Microfuge 18 Centrifuge (Beckman Coulter, Brea, CA) and the pellet was resuspended in BRB80 supplemented with 10 μM taxol (BRB80T).

**Fluorescence imaging**. The imaging was carried out in flow channels assembled from glass coverslips held together by parafilm spacers. Dichlorodimethylsilane-treated glass was passivated by 1% F127 pluronic copolymer (P2443, Sigma) for at least 30 min. Attachment of actin filaments stabilised by biotin-phalloidin was performed in channels that were incubated with an anti-biotin antibody solution (1 mg/ml in PBS, B3640, Sigma) prior to passivation. All channels were washed with 40 μl of the assay buffer upon passivation.

Actin filaments (either labelled by rhodamine or stabilised by rhodamine-phalloidin or Atto647 phalloidin) and anillin-GFP were imaged using an inverted microscope (Ti-E Eclipse, Nikon) equipped with 60x and 100×1.49 N.A. oil immersion objectives (CFI Apo TIRF and HP Apo TIRF, respectively, Nikon) in a TIRF (total internal refraction fluorescence) regime. The GFP and rhodamine fluorophores were sequentially excited by a laser on the wavelengths of 488 and 567 nm or 561 nm, respectively. FITC and TRITC fluorescence filter cubes were used, and the fluorescence was recorded using a CCD camera (iXon Ultra DU888, Andor Technology) or a CMOS camera (sCMOS ORCA 4.0 V2, Hamamatsu Photonics). The imaging setup was controlled by NIS Elements software (Nikon). The frame rate ranged between 1 frame per 10 ms to 1 frame per minute - time is indicated as time scale bars in kymographs, or as timestamps in time-lapse micrographs. The experiments were performed at room temperature.

**Anillin binding and diffusion assay**. Actin filaments stabilised by biotin-phalloidin were attached to anti-biotin antibodies on a glass coverslip. The channel was then washed with twice the channel volume of HEPES-based imaging buffer (20 mM HEPES, 2 mM MgCl₂, 1 mM EGTA, pH 7.2 KOH, 10 mM DTT, 20 mM D-glucose, 0.1 % Tween-20, 0.5 mg/ml Casein, 1 mM ATP, 0.22 mg/ml Glucose Oxidase and 0.02 mg/ml Catalase) before the HEPES-based imaging buffer with 0.12 nM anillin-GFP was flushed into the channel.

The oligomeric state of the anillin-GFP was determined from the fluorescence intensity of individual molecules diffusing on the biotin-immobilised actin filaments. This signal was compared with the fluorescence intensity of a single kinesin-1-GFP molecules immobilised on microtubules in the presence of AMP-PNP (in the absence of ATP). Kinesin-1-GFP was used as a standard due to its known dimeric state and therefore two-step photobleaching intensity profile[72]. Biotin-labelled microtubules were immobilised in the flow channel similarly to the actin filaments. The concentration of the kinesin-1-GFP was adjusted such that individual molecules bound to microtubules were detectable.

Kinetic properties of anillin-GFP unbinding from actin filaments was estimated from the temporal profile of the fluorescence intensity integrated along the contour of immobilised actin filaments upon the introduction, or washout of anillin-GFP, to or from, the imaging chamber. This process was repeated with various concentrations (from 0.12 to 12 nM) of the anillin-GFP in order to obtain a concentration profile. A complementary determination of the dissociation constant, $K_d$, was obtained from a concentration series (from 0.12 to 120 nM) of the steady-state densities of anillin-GFP on the actin filament.

**Actin filament sliding assay**. Biotin-phalloidin-stabilised actin filaments were introduced in the flow channel as described above. The channel was subsequently washed with the imaging buffer. Then, non-biotinylated actin filaments diluted in the imaging buffer containing 12 nM anillin-GFP were flushed into the channel. These filaments formed bundles with the biotin-phalloidin-stabilised filaments attached to the glass surface.

In an alternative experiment, ~1 μl of actin filaments followed by 15 μl of the imaging buffer with 0.1% methylcellulose (0.1 % w/v final concentration, 4000 cps at 2%, M0512, Sigma, supplemented with 30 mM NaCl final concentration) were first introduced into the flow channel. After initial imaging, the channel was flushed with imaging buffer containing 12 nM anillin-GFP and 0.1% methylcellulose. Filaments that remained in the channel after the flush were observed forming bundles. A control experiment was performed in the imaging buffer where ATP was replaced with 1 μM AMP-PNP, a non-hydrolysable analogue of ATP that prevents the enzymatic activity of molecular motors dependent on ATP-hydrolysis. Hexokinase (20 units/ml, Sigma H4502-1KU) was added to this buffer to remove any remnants of ATP. In this experiment, 0.5 mg/ml BSA (bovine serum albumin) was used instead of casein to block non-specific interactions.

**Actin filament disassembly assay**. Latrunculin-driven disassembly experiments were performed by introducing long unlabelled non-stabilised actin filaments into the flow channel, followed by short rhodamine-phalloidin stabilised actin filaments and 12 nM anillin-GFP in the GAB imaging buffer (GAB buffer with 10 mM DTT, 20 mM D-glucose, 0.1% Tween-20, 0.5 mg/ml Casein, 1 mM ATP, 0.8 mM PIPES, 0.22 mg/ml Glucose Oxidase and 0.02 mg/ml Catalase) supplemented with 0.1% methylcellulose and 80–200 nM latrunculin A (L12370, Thermo Fisher Scientific).

**Actin ring constriction assay**. For the ring constriction experiments, actin filaments were diluted 1:10 in the imaging buffer supplemented with 12 nM anillin-GFP and 0.1% methylcellulose. The final mixture was immediately transferred into the flow channel for imaging. We tested the robustness of the constriction under various buffer conditions. Ring constrictions were observed in both HEPES-based and GAB-based imaging buffers, with or without an additional supplement of 50 mM KCl. The data were pooled into two groups based on the properties of the actin filaments (phalloidin stabilised vs. non-stabilised) since the statistical testing (two-sample t-test at the significance level 0.05) did not reject the hypothesis that the constrictions rates within these groups come from distributions of equal means.

**Optical trapping assay**. Correlative force measurements and microscopy were performed on an optical tweezers setup equipped with confocal fluorescence imaging and microfluidic system (c-Trap, LUMICKS, The Netherlands). The continuous-wave optical tweezers were operated in a dual-trap regime. The force was measured by a position sensitive detector (PSD) in two dimensions on both beads with an acquisition rate of 50 kHz. The force was low-pass filtered to 30 Hz for storage and analysis by the controlling software (see below). The trap stiffness was calibrated using the thermal spectrum method (Scanary v. 3.3.0, LUMICKS, The Netherlands) under zero flow condition. The experiments were performed at room temperature. The force resolution of our experiment was around 100 fN at 30 Hz, with a theoretical limit of 34 fN for a typical trap stiffness of $6.55 \times 10^8$ pN/m and the corner frequency of 10.94 kHz. The system measured the distance between the beads by a built-in bright-field optical tracking with a resolution below 3 nm at 100 Hz.

The microspheres, actin filaments, imaging buffers and proteins were simultaneously flushed in the four-channel air-pressure driven microfluidic system in a laminar flow mode, ensuring that the solutions did not mix. The microfluidic system was passivated by BSA (0.1% in PBS) and F127 Pluronic no longer than 100 h before the experiment.

Actin filaments stabilised with rhodamine-phalloidin were attached to microspheres using barbed-end-binding protein gelsolin. Gelsolin-coated microspheres were prepared using a previously published protocol[35]. Briefly, the 1.01-μm-diameter carboxylated silica microspheres (SC04000, Bangs Beads) were activated by amine-reactive crosslinker chemistry using NHS (N-hydroxysuccinimide, 130672-5 G, Sigma) and functionalized by gelsolin (HPG6-A, Cytoskeleton).

To form the actin bundles (Fig. S2a), we modified a method described previously[35]. Two microspheres were captured in two traps. These microspheres were moved to the actin channel (actin filaments stabilised by rhodamine-phalloidin in the GAB imaging buffer) where ends of the filaments were attached to both of the microspheres. These constructs were then moved to the channel with 2xPBS imaging buffer (274 mM NaCl, 5.4 mM KCl, 3.6 mM $KH_2PO_4$, 16 mM $Na_2HPO_4$, 0.1 mM $CaCl_2$, 10 mM DTT, 20 mM D-glucose, 0.1% Tween-20, 0.5 mg/ml Casein, 1 mM ATP, 0.22 mg/ml glucose oxidase and 0.02 mg/ml Catalase) where we used a flow reversal for transient sideways binding of filaments to the opposite microsphere. This construct was immediately moved to the channel with 12 nM anillin-GFP in the GAB imaging buffer, where gelsolin-dependent severing of the sideway attached filaments occurred, and the anillin-GFP formed the actin bundle at the same moment. The flow in the channels was then stopped, and the force response of the construct to stretching and relaxation was measured as follows.

The microspheres were moved relative to each other in the direction of the bundle using the Trap Stepper utility of the c-trap software (v. 3.6.1, LUMICKS, The Netherlands), with 100 nm step size and 3 μm/s speed. The distance of the microspheres was measured using bright-field microscopy, and the visualisation of the actin filaments and anillin-GFP was provided by built-in scanning confocal microscope with 488 nm and 561 nm excitation lasers. The number of actin filaments in the bundle was estimated from the intensity of the fluorescence signal of the rhodamine-phalloidin. The acquisition was controlled by Scanary software (v. 3.3.0, LUMICKS, The Netherlands).

The response of the bundle to a change of the anillin-GFP concentration was measured in a five-channel microfluidic system. Similarly, as above, the actin-anillin bundle was formed and pre-stretched in 12 nM anillin-GFP channel. The construct was then moved to the channel with 1 nM anillin-GFP concentration. Due to the different geometry of the channels, the experiment was performed in the flow of about 1 μl/min to prevent mixing of the low and the high concentration anillin-GFP channels. Control experiments were performed to exclude the effects of the flow on the accuracy of the force measurement (Fig. S2f). The acquisition was controlled by Bluelake software (v 1.5, LUMICKS, The Netherlands).

**Image and data analysis**. The oligomeric state of anillin-GFP and its diffusion behaviour were quantified using FIESTA[73] for the tracking of single molecules and @msdanalyzer MATLAB tool[74] for data analysis. The background signal was calculated from the areas directly adjacent to the actin filaments or microtubules and subtracted from the fluorescence intensities for quantification. The binding kinetics of anillin-GFP was estimated from the time constant of the first-order fit to the temporal step-response concentration profiles using MATLAB (Mathworks) procedures. These profiles were measured as the anillin-GFP fluorescence intensity integrated along the contour of immobilised actin filaments upon the introduction or washout of anillin-GFP to, or from the flow chamber (see Fig. 1c). The dissociation constant was calculated from the concentration profile of the steady-state

fluorescence intensity of anillin-GFP along the contour of actin filaments using the Michaelis–Menten kinetics.

The lengths of the actin filament overlaps, as well as the circumferences of the rings, were measured manually in FiJi[75]. The circumferences of the rings were measured through the centre of the contour. Kymographs were generated using MultiKymograph plugin in FiJi or a custom-made MATLAB routine (circular kymograph in Fig. 4d). The data were analysed by self-written MATLAB scripts. Exponential decays in increasing and decreasing forms, $y = k \pm a\left(1 - e^{-\frac{t}{\tau}}\right)$ and $y = ae^{-\frac{t}{\tau}}$, respectively, were used to fit transient events. The overlaps between two filaments were identified as regions of approximately double the actin signal intensity as compared to single filaments. The velocities of sliding or disassembly were calculated as a change of the overlap length or filament length, respectively, with respect to time. Violin plots were generated using scripts by Bastian Bechtold. Boxplots were generated using MATLAB. Raw images were treated with automatic brightness/contrast enhancement routine (as implemented in FiJi) for presentation purposes. Temporal profiles of the force and distance observed in the optical trapping assay were analysed in MATLAB.

**Reproducibility and data exclusion**. All replication attempts that did not suffer from technical problems (such as improper channel passivation or malfunction of the instruments) were successful. All experiments were independently repeated at least three times, if not stated otherwise. The number of experiments, i.e. the number of individually filled channels in the case of microscopy assays and the number of constructs assembled de novo in the case of optical trapping assay, is indicated in the text.

**Reporting Summary**. Further information on research design is available in the Nature Research Reporting Summary linked to this article.

## Data availability
Source Data are available with this paper. The dataset, including the source data underlying the figures, is deposited at figshare, https://doi.org/10.6084/m9.figshare.14725188. Further information and requests for reagents can be directed to and will be fulfilled by Zdenek Lansky (zdenek.lansky@ibt.cas.cz). Source data are provided with this paper.

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

## Acknowledgements

We thank Andrew Ward for hints on actin crosslinking in the optical tweezers, the Protein Facility of MPI-CBG, Yulia Bobrova and Veronika Vanova for technical support, Karel Harant and Pavel Talacko from the Laboratory of Mass Spectrometry, BIOCEV, Faculty of Science of the Charles University, for performing the proteomic and mass

spectrometric analysis, and Laurent Blanchoin for critical reading of the manuscript. This study was supported by the projects nos. 19-27477X to Z.L. and 20-04068 S to M.B. from the Czech Science Foundation, projects 'Introduction of New Research Methods to BIOCEV' (CZ.1.05/2.1.00/19.0390) and 'BIOCEV – Biotechnology and Biomedicine Centre of the Academy of Sciences and Charles University' (CZ.1.05/1.1.00/02.0109) from the European Regional Development Fund, and by the institutional support of the Czech Academy of Sciences (RVO: 86652036). We acknowledge CMS-Biocev (Biophysical techniques) supported by MEYS CR (LM2015043), the Centre of Imaging Methods core facility, Faculty of Science, Charles University, supported by the Czech-BioImaging through MEYS CR (LM2015062 and CZ.02.1.01/0.0/0.0/16_013/0001775), and the Structural mass spectrometry core facility of CIISB, Instruct-CZ Centre, supported by MEYS CR (LM2018127). D.J. and E.P. carried out this research partly as a project work at the Faculty of Biomedical Engineering of the Czech Technical University in Prague.

## Author contributions

Conceptualisation, M.B. and Z.L.; Methodology, O.K., M.B. and Z.L.; Investigation, O.K., V.S., D.J., S.H.D. and E.P.; Formal analysis, O.K., D.J., E.P. and V.S.; Data curation, O.K., D.J., V.S. and S.H.D.; Validation, O.K.; Resources, O.K., E.Z. and S.D.; Writing, O.K., M.B. and Z.L.; Visualisation, O.K.; Supervision, M.B. and Z.L.; Funding acquisition, M.B. and Z.L.

## Competing interests

The authors declare no competing interests.
