## [Peer Review File · Nature Communications]

REVIEWER COMMENTS

Reviewer #1 (Remarks to the Author):

The authors have satisfactorily addressed my questions and I support the publication of this manuscript.

Reviewer #2 (Remarks to the Author):

This study by Kucera et al. describes the novel role of anillin in sliding actin filaments in vitro. The revised manuscript has sufficiently addressed most of the reviewer's concerns. However, the reviewer still has double in regard to the most important concern raised in the 1st round of review. The purity of the recombinant anillin and whether it contains non-myosin II remain unclear. This is despite that the authors added three lines of evidence to demonstrate that the protein prep is free of non-myosin II. As the reviewer enumerated below, all of these new results either remain unconvincing or raise fresh questions.

(1) The added SDS-PAGE gel of anillin (Fig. S1a) is not convincing. Instead, it raised more questions in regard to the purity of the prep. There are at least two major bands in the lane loaded with anillin. The major band appeared to be a protein of >200 kilodaltons (KDs). This is far larger than the predicted molecular weight (MW) of anillin-GFP at 147 KDs. The human anillin shall be of 1,087 amino acids, giving it a MW of ~120 KDs. The minor band in the lane is close to 100 KD which makes it unlikely to be anillin-GFP either. In contrary, the MW of non-muscle myosin II is actually ~ 230 KDs, closer to the major band in the lane.

(2) The added mass-spec analysis (Table S1) is confusing and not informative. The table is hard to understand without explanation. It appears that only anillin peptides was tested.

(3) The newly added experiment of blebbistatin is not conclusive (S1I and F). The concentration of this small molecule inhibitor of myosin II used by the authors was just too low. The IC50 for non-muscle myosin II is > 5 μ M, as reported by the group of James Seller and Tim Mitchison (Limouze 2004, Journal of Muscle Research and Cell Motility). The used concentration of 0.5 μ M was one order of magnitude lower than the IC50.

Lastly, the authors argued that the sliding of overlapped actin filaments proved that myosin II is not involved. The reviewer will disagree with this conclusion as the polarity of actin filaments in most experiments in this was not determined.

Another major concern that remains unresolved is the affinity between anillin and actin filaments (Fig. S1C). The measurement of this Kd has major ramifications in interpreting most experiments in which the authors used only nanomolar concentration of anillin. A similar point was raised by the reviewer 3. The presented measurement based solely on TIRF is not convincing. The best fit is still relatively poor with a R2 of only 0.25. It does not give much confidence to this data. If the amount of anillin is too slow for any classical sedimentation assay, the reviewer would suggest isothermal titration calorimetry (ITC) which uses far smaller amount of protein.

The reviewer will not recommend the acceptance of this manuscript at this point but will be happy to review it again after the two major concerns have been addressed.

Reviewer #3 (Remarks to the Author):

The authors have satisfactorily addressed my questions and concerns. I recommend publication.

We thank the reviewers for their comments. We addressed all the raised comments, and we altered the manuscript accordingly (new text is highlighted in the manuscript). Below, please find a point-by-point answer to the comments.

Reviewer #1 (Remarks to the Author):

The authors have satisfactorily addressed my questions and I support the publication of this manuscript.

We thank the Reviewer for the positive words.

Reviewer #2 (Remarks to the Author):

This study by Kucera et al. describes the novel role of anillin in sliding actin filaments in vitro. The revised manuscript has sufficiently addressed most of the reviewer's concerns. However, the reviewer still has double in regard to the most important concern raised in the 1st round of review. The purity of the recombinant anillin and whether it contains non-myosin II remain unclear. This is despite that the authors added three lines of evidence to demonstrate that the protein prep is free of non-myosin II. As the reviewer enumerated below, all of these new results either remain unconvincing or raise fresh questions. (1) The added SDS-PAGE gel of anillin (Fig. S1a) is not convincing. Instead, it raised more questions in regard to the purity of the prep. There are at least two major bands in the lane loaded with anillin. The major band appeared to be a protein of >200 kilodaltons (KDs). This is far larger than the predicted molecular weight (MW) of anillin-GFP at 147 KDs. The human anillin shall be of 1,087 amino acids, giving it a MW of ~120 KDs. The minor band in the lane is close to 100 KD which makes it unlikely to be anillin-GFP either. In contrary, the MW of non-muscle myosin II is actually ~ 230 KDs, closer to the major band in the lane.

We have performed a proteomic mass spectrometry analysis of the anillin preparation. This analysis revealed that the most abundant protein is anillin, comprising roughly 60% of the preparation. The second most abundant protein in the preparation is a heat shock protein 70, comprising roughly 4% of the preparation. All the remaining proteins are remnants comprising negligible quantities of the preparation, including myosin-II comprising about 0.2% of the preparation. Previously, we have only shown two entries from the mass spectrometry results table, comparing the amount of anillin and myosin-II in the preparation (Table S1). **Now we have added the full table as a sheet in the Excel notebook summarizing all the raw data and accompanying the manuscript.**

Additionally, we have now cut out the main band from the SDS gel and performed a MALDI-FTICR Peptide Mass Fingerprinting analysis of this band only. The result confirmed that the main band on the gel is anillin (please, see the report below for details on the sequence coverage). We speculate that the reason why anillin is migrating at an apparent larger mass size might be due to posttranslational modifications as discussed, e.g. in Yunhua et al. "Abnormal SDS-PAGE Migration of Cytosolic Proteins Can Identify Domains and Mechanisms That Control Surfactant Binding." *Protein Science*, 2012, 21 (8): 1197–1209.

(2) The added mass-spec analysis (Table S1) is confusing and not informative. The table is hard to understand without explanation. It appears that only anillin peptides was tested.

We apologize for not being clearer about the mass spectrometry results. As mentioned above, we showed only two lines from the whole mass spectrometry table with the intention to compare the relative amount of anillin and myosin-II. **We now show the whole table as a supplementary file - an Excel notebook accompanying the manuscript and summarizing all raw data gathered during the study. We also rephrased the Methods section referring to the proteomic mass spectrometry analysis to make the table clearer.**

(3) The newly added experiment of blebbistatin is not conclusive (S1I and F). The concentration of this small molecule inhibitor of myosin II used by the authors was just too low. The IC₅₀ for non-muscle myosin II is > 5µM, as reported by the group of James Seller and Tim Mitchison (Limouze 2004, *Journal of Muscle Research and Cell Motility*). The used concentration of 0.5µM was one order of magnitude lower than the IC₅₀.

We agree with the reviewer that using the inhibitor blebbistatin might not have been the best choice as different myosin motor proteins are inhibited by a given concentration of blebbistatin to a different extent. **We have now performed a new experiment to address this issue.** In the experimental setup of Fig. 1, we have exchanged the buffer including ATP for a buffer that does not contain ATP. To this buffer, we have added hexokinase to remove any remnants of ATP in the buffer and AMP-PNP (β,γ-Imidoadenosine 5'-triphosphate lithium salt hydrate), which is a non-hydrolysable analogue of ATP and which prevents the functioning of molecular motors, which are dependent on ATP-hydrolysis. At these conditions, we have repeated the experiment presented in Fig.

1g. At these conditions, in the absence of ATP and the presence of AMP-PNP, we observe robust sliding of partially overlapping filaments in the presence of anillin. The sliding is qualitatively comparable to the sliding observed in other analogous experiments presented in Fig. 1 and S1. As AMP-PNP blocks the activity of molecular motors, we conclude that the sliding is driven by the actin crosslinking protein anillin. **We present this new data in the Fig. S1i, and discuss it in the main text when describing the data of Figure 1.**

Lastly, the authors argued that the sliding of overlapped actin filaments proved that myosin II is not involved. The reviewer will disagree with this conclusion as the polarity of actin filaments in most experiments in this was not determined.

We apologize for not being clearer in this respect. We believe that the important distinction is between filament pairs that overlap fully and filament pairs that overlap partially. In our experiments, both types of filament pairs form. We thus observe some filament pairs that overlap fully and some filament pairs that overlap partially. However, we observe that only those filament pairs that overlap partially do slide and only in the direction increasing the overlap length. In other words, we never observe any directional sliding of filaments overlapping fully. Moreover, the sliding of partially overlapping filaments slows down as the overlap length increases and ceases completely once the filaments overlap fully.

If the sliding was driven by molecular motors, we would observe sliding of both types of filament pairs, fully overlapping and partially overlapping. Moreover, some of the filaments would slide in the direction decreasing the overlap length. Finally, sliding of partially overlapping filaments would not slow down as the overlap length increases. On the other hand, crosslinker generated force of an entropic origin, crucially, i) always acts in the direction increasing the overlap length, ii) decreases with the length of the overlap and iii) is zero when the filaments overlap fully. Thus, while our observations do not support the hypothesis that the sliding is driven by molecular motors, they can be fully explained by the generation of an entropic force in the system, strongly supporting the hypothesis that the sliding is driven by such a force generated by anillin. **We now reformulated the text (starting with the line 88) to make this clearer.**

Another major concern that remains unresolved is the affinity between anillin and actin filaments (Fig. S1C). The measurement of this K_d has major ramifications in interpreting most experiments in which the authors used only nanomolar concentration of anillin. A similar point was raised by the reviewer 3. The presented measurement based solely on TIRF is not convincing. The best fit is still relatively poor with a R^2 of only 0.25. It does not give much confidence to this data. If the amount of anillin is too slow for any classical sedimentation assay, the reviewer would suggest isothermal titration calorimetry (ITC) which uses far smaller amount of protein. The reviewer will not recommend the acceptance of this manuscript at this point but will be happy to review it again after the two major concerns have been addressed.

To determine the affinity of anillin for actin filaments more reliably, **we have performed a new experiment titrating increasing concentrations of anillin to the actin filaments and measuring the amount of anillin protein bound at a steady-state situation to the actin filaments.** We have visualized the anillin binding to the filaments by the TIRF microscopy method. The new results show that the K_d is 7.8 nM, the R^2 of the new fit is 0.98. **We present these new results in the Fig. S1c.** We believe that the TIRF microscopy method, which enables full visual control of the studied system, is the preferred method for studying the interactions of crosslinking proteins with filaments, as it allows to evaluate the affinity of the crosslinkers for a single species of the filament assemblies - in our case single filaments. In bulk assays (such as ITC), filaments and crosslinkers will inevitably form bundles of different sizes and other large-scale network structures, which are not well defined. The affinity value will likely differ between bundles containing various numbers of filaments. Bulk methods will thus likely yield an affinity value, which is a convolute of affinities for the various filament assemblies.

Reviewer #3 (Remarks to the Author):

The authors have satisfactorily addressed my questions and concerns. I recommend publication.

We thank the Reviewer for the positive words.

mMass Report: 210409_anillin

Date	Fri Apr 09 14:56:04 2021	Scan Number	1
Operator		Retention Time	36.34302
Contact		MS Level	1
Institution		Precursor m/z	
Instrument		Polarity	positive
		Spectrum Points	4017230
		Peak List	406

Sequence ->sp|Q9NQW6|ANLN_HUMAN Anillin OS=Homo sapiens OX=9606 GN=ANLN PE=1 SV=2

Accession	Length	Mo. Mass	Av. Mass	Coverage	Matched Int.		
	1124	125778.2939	125855.0092	64.3 %	26.3 %		
MDPFTEKLE	RTRARENLO	RKMAERPTAA	PRSMTHAKRA	RQPLSEASNO	QPLSGGEEKS	CTKPSPSKKR	CSDNTEVEVS
NLENKQPVES	TSAKSCSPSP	VSPQVOPQAA	DTISDSVAVP	ASLLGMRRGL	NSRLEATAAS	SVKTRMQKLA	EQRRRDNDND
MTDDIPESL	FSPMPSEEA	ASPPRPLLSN	ASATPVGRRG	RLANLAATIC	SWEDDVNSHF	AKQNSVQEQP	GTACLKSFSS
ASGASARINS	SSVKQEAFC	SORDGDASLN	KALSSSADDA	SLVNASISSS	VKATSPVKST	TSITDAKSC	GONPELLPKT
PISPLKIGVS	KPIVKSTLSQ	TVPSKGLSR	EICLQSQSKD	KSTTPGGTGI	KPFLERFGER	COEHSKESPA	RSTPHRTPII
TPNTKAIQER	LFKODTSSST	THLAQQLKQE	ROKELACLRG	RFDKGNISWA	EKGGNSKSKQ	LETKQETHCO	STPLKHHQGV
SKTQSLPVTE	KVTENQIPAK	NSSSTEPKGF	ECEMTKSSPL	KITLFLEEDK	SLKVTSDPKV	EQKIEVIREI	EMSVDDDDIN
SSKVINDFLES	DVLEEGELDM	EKSQEEQMDQA	LAESSEEQED	ALNISSMSLL	APLAQTVGVV	SPELSVSTPR	LELKDTSRSD
ESPKPGKFOR	TRVPRAESGD	SLGSEDRDLL	YSIDAYRSQR	EKETERPSIK	QVIVRKEDVT	SKLDEKNAF	PCOVNIKQKM
QELNNEINMQ	QTVIYQASQA	LNCVDEEHG	KGSLEEAFAE	RLLLIATGKR	TLLIDELNKL	KNEGPOKKNK	ASPOSEFMPS
KGSVTLSEIR	LPLKADFVCS	TVQKPAANY	YYLILKAGA	ENMVATPLAS	TSNSLNGDAL	TFTTFTLQD	VSNDFEINIE
VYSLVQKDP	SGLDKKKTSS	KSKAITPKRL	LTSITTKSNI	HSSVMASPGG	LSAVRTSNFA	LVGSYTLSSL	SVGNFKFVLD
KVPFLSSLG	HIYLKIKCOV	NSSVEERGL	TIFEDVSGFG	AWHRRWCVLS	GNCSISWTYP	DDEKRKNPIG	RINLANCTSR
QIEPANREFC	ARRNTFELIT	VRPQREDDRE	TLVSOCDRTL	CVTKNWLSD	TKEERDLWMO	KLNOVLVDIR	LWQPDACYKP
							IGKP

Position	Modification	Type	Mo. Mass	Av. Mass	Formula
All M	Oxidation	variable	15.9949	15.9994	O

Position	Modification	Type	Mo. Mass	Av. Mass	Formula
All C	Carbamidomethyl	fixed	57.0215	57.0514	CH ₂ CONH ₂ - H

Meas. m/z	Calc. m/z	δ (Da)	δ (ppm)	Rel. Int. (%)	z	Annotation	Formula
622.3311	622.3307	0.0004	0.6	1.11	1	[440-444] r.GRFDK.g	C27H43N9O8
628.3893	628.3889	0.0004	0.6	4.43	1	[651-655] r.TRVPR.a	C26H49N11O7
655.3523	655.3522	0.0001	0.2	2.25	1	[477-482] k.HQGVSK.t	C27H46N10O9
659.3472	659.3471	0.0001	0.1	4.43	1	[17-21] r.ENLQR.k	C26H46N10O10
674.3292	674.3290	0.0001	0.2	1.19	1	[33-38] r.SMTHAK.r	C27H47N9O9S
682.2978	682.2977	0.0000	0.1	5.02	1	[1048-1052] r.EFCAR.r [1xCarbamidomethyl]	C28H43N9O9S
684.4155	684.4151	0.0004	0.5	4.08	1	[1026-1031] r.KNPIGR.i	C29H53N11O8
690.3239	690.3239	-0.0000	-0.0	1.90	1	[33-38] r.SMTHAK.r [1xOxidation]	C27H47N9O10S
700.3375	700.3373	0.0002	0.3	1.38	1	[782-787] k.NEGPQR.k	C27H45N11O11
761.3977	761.3974	0.0002	0.3	1.49	1	[434-439] k.ELACLR.g [1xCarbamidomethyl]	C31H56N10O10S
788.3356	788.3356	0.0000	0.0	2.23	1	[381-386] r.CQEHSK.e [1xCarbamidomethyl]	C30H49N11O12S
806.4256	806.4254	0.0001	0.2	1.19	1	[696-702] r.KEDVTSK.l	C33H59N9O14
815.4485	815.4482	0.0003	0.3	3.87	1	[16-21] r.RENLQR.k	C32H58N14O11
820.4027	820.4022	0.0005	0.6	1.20	1	[1096-1101] r.DLWMQK.l	C37H57N9O10S
827.4370	827.4370	-0.0000	-0.0	12.39	1	[1041-1047] r.QJEPANR.e	C34H58N12O12
828.5552	828.5553	-0.0001	-0.2	2.86	1	[762-769] r.LLLIATGK.r	C39H73N9O10
859.4520	859.4520	0.0001	0.1	1.23	1	[888-895] k.KDPSGLDK.k	C36H62N10O14
859.4520	859.4520	0.0001	0.1	1.23	1	[889-896] k.DPSGLDK.k	C36H62N10O14
876.5400	876.5401	-0.0000	-0.0	0.64	1	[910-917] r.LLTSITK.s	C39H73N9O13
904.4527	904.4523	0.0004	0.4	2.24	1	[445-452] k.GNIWSAEK.g	C40H61N11O13
923.4682	923.4680	0.0001	0.2	0.46	1	[299-307] k.STTSITDAK.s	C37H66N10O17
928.5821	928.5826	-0.0005	-0.5	5.44	1	[327-335] k.TGVSKPIV.k.s	C42H77N11O12
934.4628	934.4629	-0.0000	-0.0	0.92	1	[1085-1092] k.NWLSADTK.e	C41H63N11O14
940.4485	940.4483	0.0002	0.3	11.96	1	[238-247] k.FSSASGASAR.i	C38H61N13O15
944.4681	944.4684	-0.0003	-0.3	3.86	1	[639-647] r.SDESPKPGK.f	C39H65N11O16
946.4844	946.4840	0.0004	0.4	1.42	1	[86-94] k.QPVESTSAK.s	C39H67N11O16
959.5159	959.5156	0.0003	0.3	2.65	1	[683-690] k.ETERPSIK.q	C40H70N12O15
961.5311	961.5313	-0.0002	-0.2	17.79	1	[802-810] k.GSVTLSEIR.l	C40H72N12O15
961.5311	961.5313	-0.0002	-0.2	17.79	1	[631-638] r.LELKDTSR.s	C40H72N12O15
976.5306	976.5310	-0.0004	-0.4	1.99	1	[134-143] r.LEATAASSVK.t	C41H73N11O16
984.5726	984.5724	0.0002	0.2	4.33	1	[397-405] r.TPIITPNTK.a	C44H77N11O14
987.5464	987.5469	-0.0006	-0.6	1.13	1	[888-896] k.KDPSGLDK.k	C42H74N12O15
987.5464	987.5469	-0.0006	-0.6	1.13	1	[889-897] k.DPSGLDK.k	C42H74N12O15
999.5463	999.5469	-0.0006	-0.6	4.25	1	[492-500] k.VTENQIPAK.n	C43H74N12O15
1002.5461	1002.5466	-0.0005	-0.5	2.50	1	[483-491] k.TQSLPVTEK.v	C43H75N11O16
1017.5509	1017.5510	-0.0001	-0.1	7.39	1	[432-439] r.QKELACLR.g [1xCarbamidomethyl]	C42H76N14O13S
1032.6414	1032.6412	0.0002	0.2	7.51	1	[909-917] k.RLLTSITTK.s	C45H85N13O14
1058.6090	1058.6092	-0.0002	-0.2	4.51	1	[771-779] r.TLLIDELNK.l	C47H83N11O16
1060.6354	1060.6361	-0.0007	-0.7	0.56	1	[899-908] k.TSKSKAITPK.r	C46H85N13O15
1069.6112	1069.6113	-0.0000	-0.0	1.29	1	[780-788] k.LKNEGPQRK.n	C45H80N16O14
1069.6363	1069.6364	-0.0001	-0.1	15.25	1	[1102-1110] k.LNQVLVDIR.l	C47H84N14O14
1090.5010	1090.5011	-0.0001	-0.1	6.87	1	[752-761] k.GSLEEAER.I	C43H71N13O20
1092.5347	1092.5354	-0.0007	-0.6	3.91	1	[351-359] r.EICLQSQSK.d [1xCarbamidomethyl]	C44H77N13O17S
1099.5669	1099.5677	-0.0008	-0.7	6.96	1	[23-32] k.MAERPTAAPR.s	C45H78N16O14S
1115.5626	1115.5626	0.0000	0.0	4.62	1	[23-32] k.MAERPTAAPR.s [1xOxidation]	C45H78N16O15S
1126.4941	1126.4946	-0.0005	-0.5	7.84	1	[255-263] k.QEATFCSQR.d [1xCarbamidomethyl]	C45H71N15O17S
1207.5358	1207.5372	-0.0014	-1.1	3.16	1	[978-987] k.CQVNSSVEER.g [1xCarbamidomethyl]	C46H78N16O20S
1208.5606	1208.5616	-0.0010	-0.8	3.54	1	[791-801] k.ASPQSEFMPSK.g	C52H81N13O18S
1214.7097	1214.7103	-0.0006	-0.5	0.91	1	[770-779] k.RTLLIDELNK.l	C53H95N15O17
1222.5180	1222.5182	-0.0002	-0.2	0.98	1	[656-667] r.AESGDSLGSSEDR.d	C46H75N15O24

Meas. m/z	Calc. m/z	δ (Da)	δ (ppm)	Rel. Int. (%)	z	Annotation	Formula
1224.5552	1224.5565	-0.0013	-1.0	3.51	1	[791-801] k.ASPQSEFMPSK.g [1xOxidation]	C52H81N13O19S
1227.6621	1227.6627	-0.0006	-0.5	4.26	1	[22-32] r.KMAERPTAAPR.s	C51H90N18O15S
1228.6198	1228.6208	-0.0010	-0.8	2.70	1	[668-677] r.DLLYSIDAYR.s	C56H85N13O18
1234.6774	1234.6790	-0.0016	-1.3	20.39	1	[681-690] r.FKETERPSIK.q	C55H91N15O17
1243.6567	1243.6576	-0.0008	-0.7	4.40	1	[22-32] r.KMAERPTAAPR.s [1xOxidation]	C51H90N18O16S
1291.6719	1291.6740	-0.0021	-1.6	0.57	1	[696-706] r.KEDVTSKLDEK.n	C54H94N14O22
1304.6401	1304.6416	-0.0014	-1.1	3.39	1	[707-717] k.NNAFPCQVNIK.q [1xCarbamidomethyl]	C56H89N17O17S
1328.6252	1328.6263	-0.0011	-0.8	1.12	1	[465-475] k.QETHCQSTPLK.k [1xCarbamidomethyl]	C54H89N17O20S
1347.6573	1347.6652	-0.0078	-5.8	3.90	1	[445-457] k.GNIWSAEKGNSK.s	C57H90N18O20
1371.6561	1371.6573	-0.0012	-0.9	3.46	1	[308-319] k.SCEGQNPPELLPK.t [1xCarbamidomethyl]	C57H94N16O21S
1456.7206	1456.7213	-0.0007	-0.5	4.76	1	[465-476] k.QETHCQSTPLK.h [1xCarbamidomethyl]	C60H101N19O21S
1473.8161	1473.8172	-0.0011	-0.7	9.29	1	[1054-1065] r.NTFELITVRPQR.e	C65H108N20O19
1507.6783	1507.6805	-0.0023	-1.5	9.28	1	[1066-1077] r.EDDRETLVQSCR.d [1xCarbamidomethyl]	C58H98N20O25S
1560.8351	1560.8380	-0.0029	-1.9	6.91	1	[362-376] k.STTPGGTGKPFLEF.f	C69H113N19O22
1589.8473	1589.8493	-0.0020	-1.3	3.10	1	[336-350] k.STLSQTVPSKGELSR.e	C66H116N20O25
1602.8861	1602.8890	-0.0029	-1.8	0.70	1	[962-975] k.VPFLSSLEGHYLK.i	C77H119N17O20
1629.9156	1629.9183	-0.0027	-1.7	5.31	1	[1053-1065] r.RNTEFELITVRPQR.e	C71H120N24O20
1644.8157	1644.8188	-0.0030	-1.8	6.73	1	[414-428] k.QDTSSTTHLAQQLK.q	C67H113N21O27
1646.7781	1646.7803	-0.0022	-1.3	2.30	1	[223-237] k.QNSVQEQPGTACLK.f [1xCarbamidomethyl]	C66H111N21O26S
1696.7191	1696.7218	-0.0027	-1.6	0.72	1	[549-563] r.EIEMSVDDDDINSSK.v	C67H109N17O32S
1712.7129	1712.7167	-0.0038	-2.2	0.66	1	[549-563] r.EIEMSVDDDDINSSK.v [1xOxidation]	C67H109N17O33S
1737.7561	1737.7596	-0.0035	-2.0	0.49	1	[71-85] r.CSDNTEVEVSNLENK.q [1xCarbamidomethyl]	C68H112N20O31S
1769.8939	1769.8963	-0.0024	-1.4	0.44	1	[918-935] k.SNIHSSVMASPGGLSAVR.t	C73H124N24O25S
1785.8879	1785.8912	-0.0033	-1.8	9.54	1	[918-935] k.SNIHSSVMASPGGLSAVR.t [1xOxidation]	C73H124N24O26S
1789.8857	1789.8901	-0.0045	-2.5	0.53	1	[703-717] k.LDEKNNAFPCQVNIK.q [1xCarbamidomethyl]	C77H124N22O25S
1803.9563	1803.9599	-0.0036	-2.0	6.43	1	[360-376] k.DKSTTPGGTGKPFLEF.f	C79H130N22O26
1862.0201	1862.0243	-0.0041	-2.2	1.34	1	[180-198] k.AASPPRPLLSNASATPVGR.r	C80H136N26O25
1898.9044	1898.9090	-0.0046	-2.4	2.56	1	[42-59] r.QPLSEASNQQPLSGGEEK.s	C78H127N23O32
2008.9973	2009.0033	-0.0060	-3.0	3.55	1	[272-292] k.ALSSSADDASLVNASISSVK.a	C82H141N23O35
2058.0175	2058.0210	-0.0035	-1.7	0.60	1	[414-431] k.QDTSSTTHLAQQLKQER.q	C83H140N28O33
2126.0408	2126.0473	-0.0065	-3.1	3.02	1	[40-59] r.ARQPLSEASNQQPLSGGEEK.s	C87H144N28O34
2146.0958	2146.1026	-0.0069	-3.2	0.39	1	[936-956] r.TSNFALVGSYTLSSVGNK.f	C94H152N24O33
2195.0372	2195.0424	-0.0052	-2.4	0.47	1	[564-582] k.VINDLFSVDLEEGELDMK.s	C95H151N21O36S
2211.0304	2211.0373	-0.0069	-3.1	2.22	1	[564-582] k.VINDLFSVDLEEGELDMK.s [1xOxidation]	C95H151N21O37S
2282.1394	2282.1484	-0.0090	-3.9	0.53	1	[39-59] k.RARQPLSEASNQQPLSGGEEK.s	C93H156N32O35
2432.1149	2432.1212	-0.0063	-2.6	6.90	1	[656-677] r.AESGDSLGSSEDRLLYSIDAYR.s	C102H158N28O41
2665.2197	2665.2257	-0.0060	-2.3	0.67	1	[71-94] r.CSDNTEVEVSNLENKQPVVESTAK.s [1xCarbamidomethyl]	C107H177N31O46S
2692.3602	2692.3835	-0.0233	-8.7	0.66	1	[460-482] k.QLETKQETHCQSTPLKKGQVSK.t [1xCarbamidomethyl]	C113H190N36O38S
2692.3602	2692.3691	-0.0089	-3.3	0.66	1	[815-837] k.ADFVCSTVQKPAANYYLIILK.a [1xCarbamidomethyl]	C125H190N28O36S
3368.6248	3368.6481	-0.0233	-6.9	0.64	1	[976-1004] k.IKCQVNSSVEERGLTIFEDVSGFGAWHR.r [1xCarbamidomethyl]	C150H226N42O45S
3368.6248	3368.6461	-0.0213	-6.3	0.64	1	[95-127] k.SCSPSPVSPQVQQAADTISDSVAVPASLLGMR.r [1xCarbamidomethyl; 1xOxidation]	C141H234N40O51S2

REVIEWERS' COMMENTS

Reviewer #2 (Remarks to the Author):

The authors have satisfactorily addressed all of my concerns in this round of revision. The reviewer would recommend the publication of this manuscript.

The reviewer would highly recommend the authors to move Figure S1A-C to be part of the main figures. It demonstrated the purify of recombinant anillin as well as its binding of actin filaments. Both are significant advances that would be greatly appreciated by the researchers in the field.

Reviewer #2 (Remarks to the Author):

The authors have satisfactorily addressed all of my concerns in this round of revision. The reviewer would recommend the publication of this manuscript.

The reviewer would highly recommend the authors to move Figure S1A-C to be part of the main figures. It demonstrated the purify of recombinant anillin as well as its binding of actin filaments. Both are significant advances that would be greatly appreciated by the researchers in the field.

We thank the Reviewer for the positive words. We agree with the proposed moving of panels regarding anillin purity and actin-binding properties from supplementary figures to the main figures. They now appear as Fig. 1 b-e.